# Processing, Phase Stability, and Conductivity of Multication-Doped Ceria

**Elizabeth Gager *** [ID] **and Juan C. Nino**

Department of Materials Science and Engineering, University of Florida, Gainesville, FL 32611, USA; jnino@mse.ufl.edu
* Correspondence: egager@ufl.edu

**Abstract:** Multicomponent doping of ceria with four cations is used as a preliminary investigation into the ionic conductivity of high-entropy-doped ceria systems. Different compositions of $Ce_{1-x}(Nd_{x/4}Pr_{x/4}Sm_{x/4}Gd_{x/4})O_{2-\delta}$ (x = 0.05, 0.10, 0.15, and 0.20) are synthesized using the oxalate co-precipitation method yielding single-phase oxalate precursors. X-ray diffraction, Raman spectroscopy, and Fourier-transform infrared spectroscopy are used to characterize the precipitated oxalates. Simultaneous thermal gravimetric analysis and differential scanning calorimetry reveal a two-step decomposition of the oxalates into the doped oxide. The ionic conductivity of the samples is measured from 250 °C to 600 °C using electrochemical impedance spectroscopy. All samples exhibit similar grain conductivity values at 600 °C, comparable to singly doped samples. However, an increase in total conductivity is observed with an increase in doping concentration up to 15% followed by a decrease beyond this concentration. These findings suggest that multicomponent doping may not significantly enhance the grain conductivity of doped ceria beyond conventional single and co-doped compositions but can modulate the grain boundary conductivity and thus the total conductivity of ceria ceramics.

**Keywords:** doped ceria; oxalate method; multi-component oxides; ionic conductivity





## 1. Introduction

Solid oxide fuel cell (SOFC) technology offers efficient electricity generation from a wide variety of fuels, making it very useful for power generation applications [1]. Advancement in SOFC research has led to the identification of doped cerium oxide materials as potential candidates [2]. Yttria-stabilized zirconia (YSZ)-based SOFCs require high operating temperatures (700 °C to 1000 °C) to achieve the necessary performance [1,3]. Although doped ceria exhibits some instability at high temperatures due to electronic conduction resulting from the reduction of $Ce^{4+}$ to $Ce^{3+}$, the doped cerium oxide materials are able to achieve superior ionic conductivity in the 400 °C to 700 °C temperature range [4]. Therefore, they offer the ability to lower the operating temperature, reducing costly materials and large energy inputs. Cerium oxide is generally singly or co-doped with 10% to 20% trivalent rare earth dopants, and a maximum in conductivity is generally observed at a 10% dopant concentration [5]. However, concentrations up to 20% have been shown to increase the conductivity depending on the dopants.

The conductivity of singly doped ceria systems is well documented for the different rare earth elements [6,7]. Samarium and gadolinium have shown superior conductivity compared to other doped ceria systems. However, limitations in the improvement of conductivity are observed due to interactions between the dopant and vacancies, which create defect associates that restrict the movement of oxygen within the lattice. Andersson et al. showed that the association energy can be minimized by doping with Pm [8]. Alternatively, a co-doped system can be used to tailor the combination of electrostatic and elastic energies within the system to break up defect associates within the material and

maintain increased conductivity with further doping. Specifically, a combination of Nd/Sm or Pr/Gd was identified to achieve similar results to doping with Pm. Omar et al. showed that co-doping ceria with Nd and Sm significantly improved the ionic conductivity compared to gadolinium-doped ceria [9]. Additionally, it was shown that 15% doping had the highest conductivity, compared to singly doped systems that generally show a maximum at 10% doping. While co-doping has been explored and shown improvements in conductivity, doping with more than two elements is rarely explored for ionic conductivity [10]. Doping with more than two elements begins to increase the number of secondary phases that may be present in the material, leading to difficulties in processing.

The synthesis of complex multicomponent oxides can be performed using a variety of methods, but it is typically understood that liquid-phase methods will lead to better homogeneity. Conversely, typical solid-state synthesis, which is often used for doped ceria compounds, can lead to segregation in the material resulting from inadequate mixing. In addition, solid-state methods typically lead to large particle sizes of calcined powder and require a higher sintering temperature, which may result in the segregation of dopants at the grain boundary.

Among the liquid-phase synthesis methods (e.g., precipitation, hydrolysis, citrate gel, etc.), co-precipitation using oxalic acid as the precipitating agent is a particularly effective method to reduce segregation [11]. The oxalate method produces nanocrystalline single-phase precipitates that are already mixed for the lanthanide elements [12]. All lanthanide oxalates form in the $P2_1/c$ space group, allowing a single-phase precipitate to be formed during the synthesis method. This method has been shown to be very effective for processing complex doped ceria samples with maintained homogeneity throughout the heat treatment [12].

At the turn of the century, high-entropy alloys (HEAs) or multicomponent alloys, were first proposed to explore the stabilization of a single phase with configurational entropy [13,14]. These alloys contain five or more principal elements in near equal atomic percent and show potential for a wide range of applications from functional to structural materials [15]. More recently, research has expanded the idea of high entropy to the class of oxide ceramics. In 2015, Rost et al. showed that an equimolar concentration of Mg, Ni, Zn, Cu, and Co could be reversibly stabilized into a single-phase system [16]. The space of high-entropy ceramics has expanded with applications in wear-resistant coatings, water splitting, and more as the compositional space grows drastically with the increase in the number of dopants [17]. The large number of combinations makes these particularly interesting to tailor the properties of materials. Additionally, the random orientation within the structure of the material may offer benefits for ionic conductivity and the migration of oxygen through the system. The introduction of randomness throughout the material may break up defect associates that restrict the motion of oxygen through the system. Here, we explore multicomponent doping within the ceria system using Pr, Nd, Sm, and Gd from 5% to 20% doping.

## 2. Results and Discussion

### 2.1. Metal Oxalate Characterization

Figure 1 displays the XRD patterns for the synthesized oxalate precursors. All peaks are assigned to the metal oxalate with no secondary phases present. Specifically, all synthesized samples form in the $P2_1/c$ space group as decahydrate oxalates. Since all lanthanides form with the same space group, a single-phase oxalate structure is precipitated during multication synthesis rather than a mixture of individual metal oxalates. An increase in doping concentration leads to a shift in the pattern to higher 2θ values, representing a decrease in the lattice parameters. This can specifically be seen in the (111) peak in Figure 1b. All of the metal cations sit in 9-fold coordination with a +3 charge. The ionic radius of the cations decreases from cerium at 1.196 Å to 1.107 Å for gadolinium. As the doping increases, the average ionic radius within the material decreases, thus shifting the (111) peak to higher 2θ values and shrinking the a and c lattice parameters. The changes in lattice

parameters follow the same trend as observed by Alemayehu et al. for gadolinium-doped cerium oxalate and are recorded in Table 1 [12]. As seen, the a and c lattice parameters shrink while b and β stay relatively constant, and a reduction in volume is observed. This is consistent with previous work on lanthanide oxalates that showed a decrease in the a and c parameters with an increase in atomic number through gadolinium for the lanthanides and only a slight change in b and β [18]. The low doping concentrations used here prevent a noticeable trend from being observed for the b and β parameters.

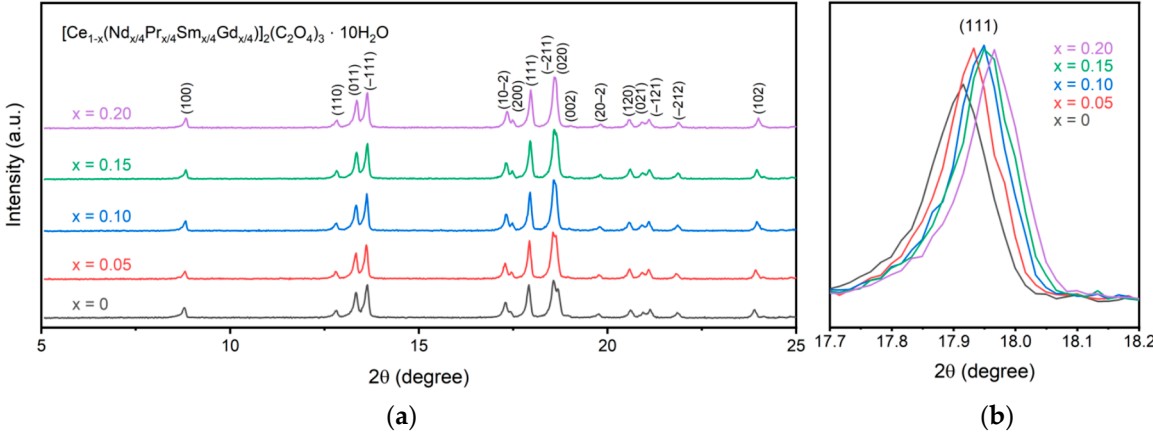

**Figure 1.** XRD patterns of the synthesized oxalate precursors (**a**) with a comparison of the (111) peak (**b**) showing a shift to smaller lattice parameters for higher dopant concentrations.

**Table 1.** Lattice parameters, a, b, c, and β, and unit cell volume of the $[Ce_{1-x}(Pr_{x/4}Nd_{x/4}Sm_{x/4}Gd_{x/4})]_2(C_2O_4)_3$ 10 $H_2O$ samples.

| Sample | a (Å) | b (Å) | c (Å) | β (°) | Volume (Å³) |
|---|---|---|---|---|---|
| x = 0 | 11.32 | 9.63 | 10.42 | 114.5 | 1037 |
| x = 0.05 | 11.30 | 9.63 | 10.39 | 114.5 | 1033 |
| x = 0.10 | 11.29 | 9.62 | 10.37 | 114.5 | 1026 |
| x = 0.15 | 11.27 | 9.62 | 10.35 | 114.5 | 1023 |
| x = 0.20 | 11.26 | 9.63 | 10.34 | 114.4 | 1021 |

Raman and Fourier-transform infrared spectroscopy can be coupled together to investigate the bonding nature within a material. Three distinct bands are typically observed in Raman spectroscopy of the lanthanide oxalates: $v_{CO}$ at ~1480 cm$^{-1}$, corresponding to the symmetric stretching of the C–O single and double bonds; $v_{CC}$ at ~920 cm$^{-1}$, the stretching of the C–C single bond; and $\delta_{OCO}$ at ~500 cm$^{-1}$, associated with the bending of the O–C–O bond [19,20]. The $\delta_{OCO}$ band at ~500 cm$^{-1}$ contains a greater contribution from the metal–oxygen stretching motion and, therefore, will display larger shifts compared to the other bands identified with Raman spectroscopy [21]. While Tamain et al. did not observe a shift between the Pr, Nd, and Sm oxalates for either the $v_{CO}$ or $\delta_{OCO}$ bands, they did observe a shift for Gd [19]. Morris and Hobart, however, observed a shift to higher wavenumbers with increasing atomic numbers for all rare earth oxalates up to Er [21]. Raman spectroscopy for the synthesized oxalates is shown in Figure 2a, and the recorded values for the $v_{CO}$, $v_{CC}$, and $\delta_{OCO}$ bands are recorded in Table 2. The most intense band is observed for all oxalates around 1475 cm$^{-1}$ for the $v_{CO}$ band. The shift observed for this band is less than a wavenumber and significantly less than what is observed for gadolinium oxalate. The dopant concentration may be too low to significantly alter the bonding environment, causing only very small shifts. This is also similar to results obtained by Tamain et al., who did not observe any shift for Pr, Nd, or Sm oxalates compared to Ce oxalates [19]. Additionally, no doublet is observed for the $v_{CO}$ band at 1475 cm$^{-1}$. A doublet was previously reported by Tamain et al. and Clavier et al. and was described with

two distinct oxalate ions in the unit cell [19,22]. The lack of a doublet further confirms the single-phase purity of the oxalate rather than a mixture of individual oxalates. A small shift in the $\nu_{CO}$ band to a higher wavenumber is observed with increased doping concentration. This is in line with previously published results that showed smaller dopants lead to a shift in the spectrum to higher wavenumbers [18,19]. Additionally, a peak around 980 cm$^{-1}$ is observed for all doped samples, which is not observed in the individual oxalates. This band was previously identified using Raman spectroscopy by Tamain et al. and is not visible in FTIR [19].

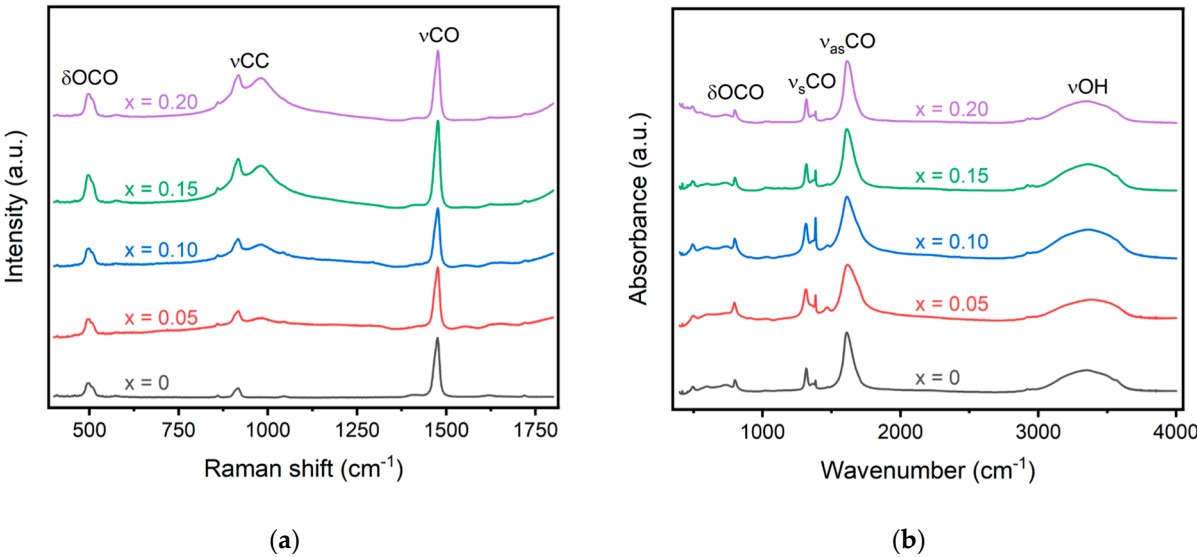

(**a**)          (**b**)

**Figure 2.** (**a**) Raman and (**b**) FTIR spectra of the oxalate precursors. Significant bands corresponding to O-C-O, C-C, C-O, and O-H bonds are displayed in the figure.

**Table 2.** Raman and FTIR wavenumbers for the various oxalate samples. All values are given in cm$^{-1}$.

| Spectroscopy | Vibration Mode | x = 0 | x = 0.05 | x = 0.10 | x = 0.15 | x = 0.20 |
|---|---|---|---|---|---|---|
| Raman | $\nu_{CO}$ | 499 | 500 | 500 | 500 | 501 |
| | $\nu_{CC}$ | 915 | 914 | 913 | 913 | 913 |
| | $\delta_{OCO}$ | 1473 | 1474 | 1474 | 1474 | 1475 |
| | $\nu_{CO}$ | 1610 | 1614 | 1613 | 1612 | 1613 |
| FTIR | $\nu_{CO}$ | 1317 | 1315 | 1314 | 1317 | 1318 |
| | | 1360 | 1359 | 1362 | 1363 | 1363 |
| | $\delta_{OCO}$ | 802 | 799 | 800 | 802 | 801 |

In infrared spectroscopy, the $\nu_{CO}$ band at ~1480 cm$^{-1}$ is forbidden, and the $\nu_{CC}$ band at ~920 cm$^{-1}$ is almost invisible. While many of the bands observed in the IR spectra are not sensitive to variation in the metal cation, the $\nu_{CO}$ band at ~1600 cm$^{-1}$ increases with an increase in atomic radii. While this band is observed to increase with the introduction of dopants for the oxalate compounds, a consistent increase is not observed with an increase in doping. The $\nu_{CO}$ band in undoped cerium oxalate is observed at 1610 cm$^{-1}$, and a 6 cm$^{-1}$ increase is observed with 5% doping. Upon further doping to 10%, the $\nu_{CO}$ band decreases to 3 cm$^{-1}$ and remains steady with an increase in doping concentration. This may signal inhomogeneity within the sample or could be a result of different amounts of structural water within the material, thus altering the bond structure. Large changes in the wavenumber of the bands, however, are not expected within the materials for such small substitution levels. The prior literature only observed a shift of ~4 cm$^{-1}$ between pure Pr and Gd oxalate [19].

The thermal decomposition of individual rare earth oxalates is well-recorded in the literature [23–26]. Depending on the element, the decomposition can proceed through

dehydration and then decomposition of the oxalate directly into the oxide or through a carbonate intermediate. During the dehydration step, intermediate oxalates may occur with less water if there is a non-equivalency in the water molecule bond strength [25]. This was observed for Gd, Sm, and Nd oxalates depending on the initial water concentration in the molecule. Dehydration of Ce and Pr oxalates in air proceeds in one step. Notably, the Ce oxalate decomposition temperature is well below the other rare earth oxalates at ~300 °C compared to ~450 °C for Nd. The decomposition of the oxalate proceeds through the breaking of the RE-O bond [25]. The bond length decreases with an increase in the atomic number. Therefore, as the doping concentration increases, the decomposition temperature will increase due to the substitution of cerium for elements with smaller radii and bond lengths. While some rare earth oxalates decompose through a dioxycarbonate intermediate, it has been previously shown that multicomponent oxalates undergo non-additive decomposition behavior and thus are not expected to form intermediates [24]. Since the maximum doping concentration is 20%, it is expected that the decomposition of the oxalates will behave more like cerium oxalate and only have two decomposition steps.

The decomposition of the oxalate precursors is shown in Figure 3 for both the change in weight and heat flow of the materials. The onset temperatures, enthalpies, and mass changes for each decomposition step are recorded in Table 3. All oxalates decompose in a two-step process, with the first step being the dehydration of the oxalate followed by the conversion of the oxalate into the oxide with the release of $CO_2$. Even though the oxalates contain rare earth elements other than Ce, they do not show conversion to an intermediate phase like pure rare earth oxalates other than Ce do [23]. The lack of an intermediate phase further confirms that the multication-doped oxalate forms in a single-phase solution rather than a solution of mixed oxalates. The dehydration onset temperature increases slightly with the dopant concentration from 102 °C for pure cerium oxalate to 108 °C for the 10% and 20% doped cerium oxalate but does not follow a direct trend in increasing temperature with increasing dopant. A direct comparison between the dopant concentration and dehydration temperature is not expected due to freely arranged water molecules in the cell that does not depend on metal cation [12]. Additionally, the calculated mass loss for the doped oxalates is ~9.7 for all compositions, in good agreement with the theoretical value of 10 water molecules.

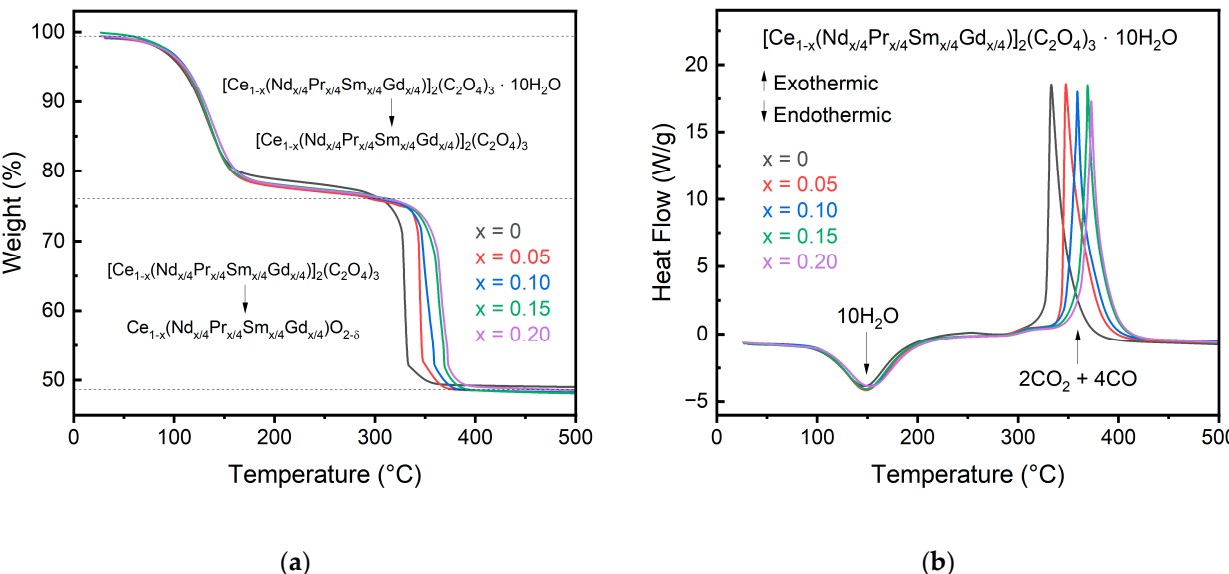

**(a)** **(b)**

**Figure 3.** SDT data for the oxalate precursors showing the change in (**a**) weight and (**b**) heat flow during oxalate decomposition. A slight shift in the decomposition temperature to higher temperature is observed for greater substitution concentrations.

**Table 3.** Onset temperatures, enthalpies, and mass changes for each of the two steps in the decomposition reaction of the doped cerium oxalates.

| Sample | $RE_2(C_2O_4)_3 \cdot xH_2O \rightarrow RE_2(C_2O_4)_3$ | | | | $RE_2(C_2O_4)_3 \rightarrow REO_{2-\delta}$ | | |
| | Temperature (°C) | Enthalpy (kJ/mol) | Mass Change (%) | $xH_2O$ (moles) | Temperature (°C) | Enthalpy (kJ/mol) | Mass Change (%) |
|---|---|---|---|---|---|---|---|
| x = 0 | 101.6 | 341 | 22.5 | 8.77 | 326.9 | −441 | 28.6 |
| x = 0.05 | 104.5 | 347 | 24.6 | 9.87 | 341.1 | −489 | 27.3 |
| x = 0.10 | 108.1 | 323 | 24.4 | 9.78 | 343.5 | −476 | 27.6 |
| x = 0.15 | 105.0 | 334 | 24.2 | 9.69 | 355.5 | −497 | 28.0 |
| x = 0.20 | 108.1 | 328 | 24.2 | 9.70 | 358.9 | −488 | 27.7 |

Following dehydration, the oxalate begins decomposition around 327 °C for pure cerium. The decomposition temperature for the oxalate is increased as the dopant concentration increases. This can be explained with the hard–soft acid–base theory, where the dopants (Pr, Nd, Sm, Gd) will have a higher charger density than Ce, forming a more stable complex and delaying decomposition [12]. The decomposition onset temperature for the 20% doped cerium oxalate is 359 °C, more than a 20 °C increase for just 20% doping. All oxalate samples complete decomposition to their respective oxide by 400 °C compared to individual Pr, Nd, Sm, and Gd oxalates, which do not begin to decompose until after 400 °C and finish around 700 °C.

*2.2. Doped Cerium Oxide Phase Analysis*

Pure $CeO_2$ forms in the fluorite $Fm\bar{3}m$ space group. Doping with trivalent elements causes a transition to lower symmetry structures such as the C-type bixbyite structure at higher doping concentrations. The doping concentration of 20% is lower than the solubility in the fluorite structure that was found for the individual dopants; thus, the fluorite structure is expected to be formed for all experimental compositions. Figure 4 displays the XRD patterns for the calcined powders. As shown, all samples form in the fluorite $Fm\bar{3}m$ space group. Similar to the oxalate powders, a shift is observed in the peak positions between the different compositions. The shift, however, is to lower 2θ values, occurring from an increase in the ionic radius of the dopants compared to cerium. Since cerium is assumed to have a +4 charge, as no reducing atmosphere was used, the lower trivalent charge of the dopants leads to an increase in the average ionic radius. A comparison of the (111) peak is shown for the compositions in Figure 4b. While the variation in the lattice parameter is only about 0.02 Å, a clear trend is observed, nonetheless. The lattice parameter linearly increases with increasing dopant concentration, following Vegard's law [27]. The short-range ordering of oxygen vacancies may be less pronounced than in singly doped ceria systems that do not follow Vegard's law [9]. The linear relationship is obtained between the doping concentration, x, and the lattice parameter as displayed in Equation (1).

$$a_o\,(x) = 0.08598x + 5.4107 \tag{1}$$

The lattice parameter for pure ceria is found to be 5.411 Å, while that of the 20% doped concentration is found to be 5.427 Å. This is in line with what is expected based on singly doped ceria systems for Pr, Nd, Sm, and Gd. At 20% doping, Pr has a lattice parameter of 5.409 Å, Nd of 5.443 Å, Sm of 5.433 Å, and Gd of 5.423 Å [28–31]. The 5.427 Å experimental lattice parameter is the average of the four singly doped compositions, following the rule of mixtures.

Additionally, the crystallite size was calculated for the samples between 400 °C and 800 °C, shown in Figure 4c. The crystallite size decreases with increasing dopant concentration. As described by Alemayehu et al. and seen in Figure 3, an increase in dopant concentration increases the decomposition temperature, leading to a delay in the oxide crystallization [12]. At 400 °C, the crystallite size is approximately 5 nm for all compositions. This increases to 23 nm for the 20% doped sample compared to 33 nm for pure ceria.

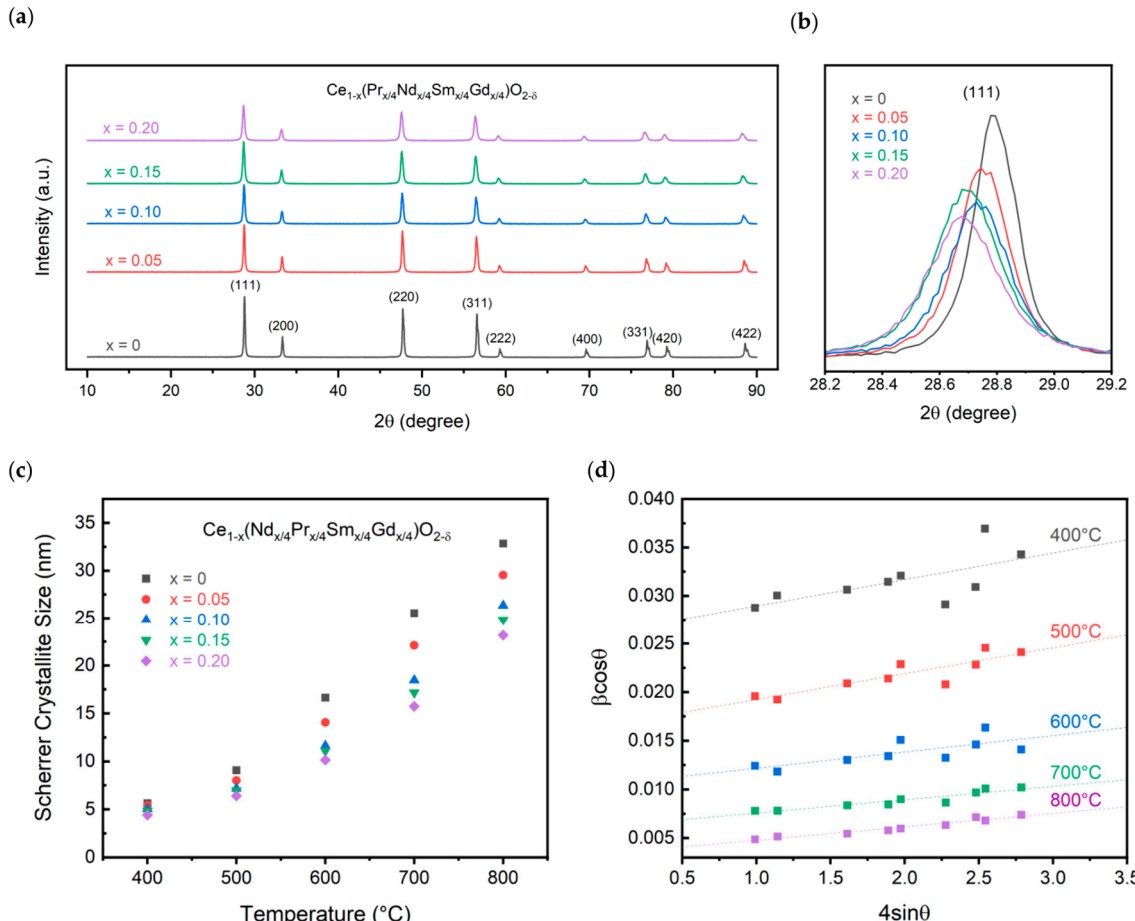

**Figure 4.** XRD characterization showing (**a**) XRD patterns, (**b**) (111) peak comparison, and (**c**) crystallite size for all calcined compositions and (**d**) Williamson-Hall plot for the 20% substitution sample.

The effect of size and strain on the system was evaluated and shown in the Williamson–Hall plot in Figure 4d for the 20% doped sample. At a calcination temperature of 400 °C, the powders do not display a linear relationship that can be used to calculate the size and strain within the particles. Upon increasing the temperature to 800 °C, a linear relationship is observed with a strain calculated to be $1.38 \times 10^{-3} \pm 1.22 \times 10^{-4}$. Interestingly, the calculated strain in the system decreases with increasing dopant concentration. CeO$_2$ calcined at 800 °C has a calculated strain value of $1.73 \times 10^{-3} \pm 5.29 \times 10^{-5}$. While the crystallite size is larger than that calculated using Scherrer's equation at 41 nm compared to 23 nm, the trend holds that increasing the dopant concentration decreases the size.

Raman spectroscopy can investigate the local structure and disorder of the material in contrast to XRD, which investigates the long-range order of the structure. In pure CeO$_2$, only one mode, F$_{2g}$, is active and centered at ~460 cm$^{-1}$, although two additional modes centered around 250 cm$^{-1}$ and 600 cm$^{-1}$ may be observed due to defects [32]. The F$_{2g}$ breathing mode corresponds to the CeO$_8$ octahedra with a central stationary Ce atom surrounded by 8 O$^{2-}$ ions. In the cubic RE$_2$O$_3$ structure, 22 active Raman modes are predicted. However, due to the low intensity, not all modes are able to be identified, and most focus on the F$_g$ mode centered at ~350 cm$^{-1}$. Upon doping CeO$_2$ with trivalent cations, changes in the F$_{2g}$ mode, the introduction of oxygen vacancy modes, and the introduction of the F$_g$ mode of the lower symmetry cubic RE$_2$O$_3$ phase are tracked. Doping with trivalent cations, which introduce vacancies into the structure, will cause a reduction in the average coordination number of Ce, causing a slight red shift in the F$_{2g}$ mode. As the dopant concentration increases, the red shift will increase until high doping concentrations, at which the red shift is offset by a blue shift, caused by the expansion of the lattice. Additionally, F$_{2g}$ broadens,

decreases in intensity, and becomes progressively more asymmetric with increasing RE doping due to alterations in the surrounding oxygen ions [30]. A peak corresponding to oxygen vacancies can be seen in doped ceria systems around $570 \text{ cm}^{-1}$ and is ascribed to oxygen vacancies that form to maintain charge neutrality when trivalent dopants are introduced [33,34]. A band at $600 \text{ cm}^{-1}$ that may be observed is instead ascribed to intrinsic vacancies due to the presence of $Ce^{3+}$ rather than other rare earth dopants [35]. In a trivalent-doped ceria system, if the fluorite structure is maintained, no peak should be observed at $\sim 350 \text{ cm}^{-1}$, which corresponds to the lower symmetry C-type structure.

Raman spectroscopy for the calcined oxides is shown in Figure 5, and the fitted values are shown in Table 4. All samples show the $F_{2g}$ peak corresponding to the fluorite structure with a second peak corresponding to oxygen vacancies seen in all doped samples. The $F_{2g}$ peak shifts to a slightly lower wavenumber with increasing dopant concentration from $462 \text{ cm}^{-1}$ for undoped ceria to $455 \text{ cm}^{-1}$ for 20% doped ceria. The shift to lower energy is a result of an increase in lattice strain due to the size mismatch between the dopants and the host atoms. The mismatch causes a reduction in energy of the Ce-O vibrations, shifting the band to a lower wavenumber [32]. The $F_{2g}$ peak also shifts to a lower wavenumber as a result of the reduction in the average coordination number with an increase in trivalent dopants and thus oxygen vacancies. The small crystallite size may also influence the shift to lower wavenumbers where a red shift was documented for nanosized $CeO_2$ [36]. The calculated FWHM increases with increasing doping concentration for the $F_{2g}$ band as expected due to alterations in the surrounding oxygens, disrupting the symmetry of the $CeO_8$ octahedra. Additionally, a second peak is observed in the spectra for all doped samples around $570 \text{ cm}^{-1}$, resulting from the formation of oxygen vacancies. The oxygen vacancy peak shifts to a higher wavenumber from $569 \text{ cm}^{-1}$ to $573 \text{ cm}^{-1}$ for the 5% and 20% doped materials, respectively. The increase in FWHM for the longitudinal oxygen vacancy band (LO) and shift towards $600 \text{ cm}^{-1}$ may signify the introduction of $Ce^{3+}$ in the system in addition to the trivalent dopants [35]. The A band around $360 \text{ cm}^{-1}$, corresponding to the lower symmetry bixbyite structure, is not observed in the Raman spectra, confirming the lack of a secondary lower symmetry phase. Additionally, the oxygen vacancies can be estimated based on the FWHM, the ratio of peak intensities, and the ratio of peak areas [34]. The values obtained after fitting the spectra with Origin are shown in Table 4. Comparing the intensities of the peaks, an increase in the doping concentration increases the number of oxygen vacancies, which is expected due to the increase in trivalent cations. However, comparing the areas of the peaks, a maximum is observed for the 15% doped ceria. Additionally, the vacancy concentration is much higher than expected based on the stoichiometry of the material. This may be a result of the laser wavelength that was used. The 532 nm laser has a smaller penetration depth than a 785 nm laser and may be more representative of the oxygen vacancies at the surface rather than the bulk of the material [35]. Overall, a significant number of oxygen vacancies were introduced into the material while still maintaining the fluorite structure.

Scanning electron microscopy (SEM) and energy dispersive spectroscopy (EDS) were performed on the sintered pellets, as shown in Figure 6. The calculated grain size for the samples ranges from 1.2 μm to 1.5 μm. Using the oxalate method, the required sintering temperature is much lower than for the solid-state method, thus allowing for the synthesis of pellets with smaller grain sizes. The elements appear to be well distributed throughout the material, and no dopant segregation along the grain boundaries appears to be present at this scale. The 15% doped sample does show some areas with relatively lower Ce concentration in the EDS scan, although this may be attributed to the small pores and uneven surface of the pellets. Due to the similar L$\alpha$ energy levels and small dopant concentrations that lead to large errors within EDS calculations, individual doping values are not recorded here. However, the total calculated dopant concentration from EDS is within 15% of the expected value for each of the samples, with the 5% doped sample having the largest difference between the expected and calculated value.

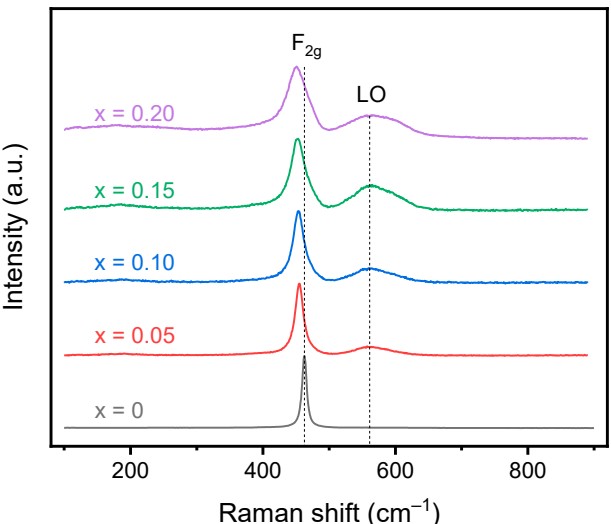

**Figure 5.** Raman spectroscopy data for the calcined doped cerium oxides. A red shift in the $F_{2g}$ peak is observed with increasing dopant concentration as a result of the increase in lattice strain. The appearance of the LO peak is observed for the doped ceria compositions resulting from oxygen vacancies. No peak corresponding to the C-type structure is observed.

**Table 4.** Raman spectroscopy values for the $F_{2g}$ and LO peak location, full width at half max (FWHM), and ratios of the peaks.

| Dopant Concentration | $F_{2g}$ (cm$^{-1}$) | FWHM $F_{2g}$ (cm$^{-1}$) | LO (cm$^{-1}$) | FWHM LO (cm$^{-1}$) | $I_{570}/I_{460}$ | $A_{570}/A_{460}$ |
|---|---|---|---|---|---|---|
| x = 0 | 462.50 | 10.25 | - | - | - | - |
| x = 0.05 | 459.24 | 19.08 | 569.45 | 60.89 | 0.22 | 0.40 |
| x = 0.10 | 458.06 | 24.88 | 571.02 | 57.05 | 0.32 | 0.40 |
| x = 0.15 | 457.03 | 32.37 | 574.09 | 69.80 | 0.32 | 0.54 |
| x = 0.20 | 454.68 | 43.40 | 572.89 | 74.07 | 0.44 | 0.49 |

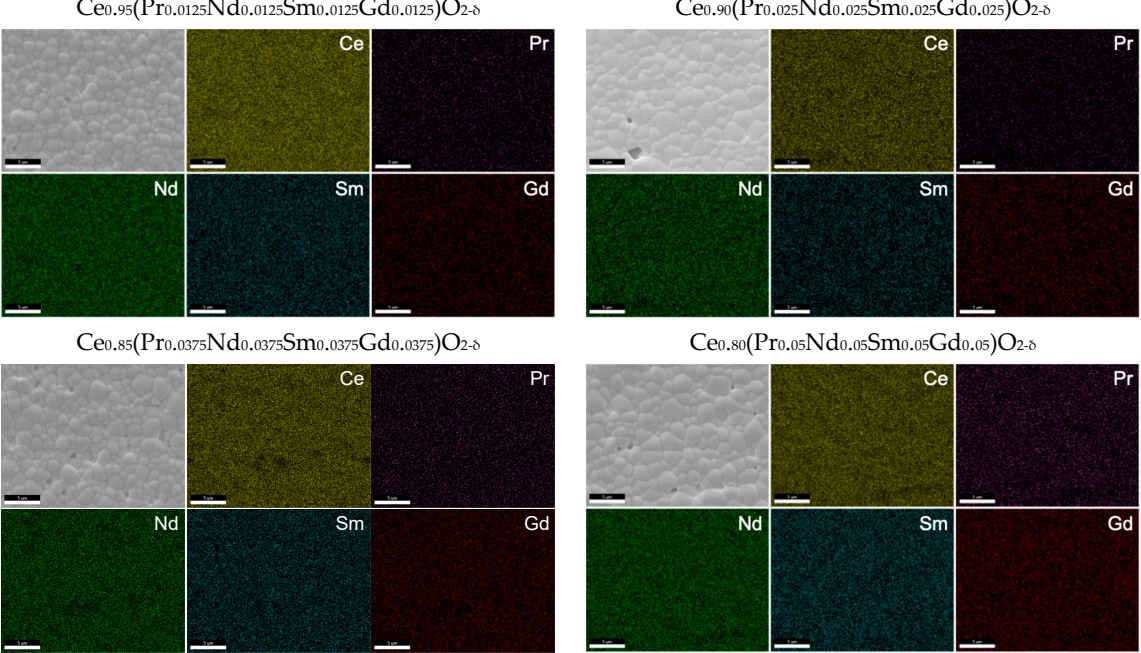

**Figure 6.** SEM and EDS of the sintered pellets used for conductivity measurements. All samples display homogeneous chemical composition and dense microstructure with no abnormal grain growth.

### 2.3. Ionic Conductivity

The ionic conductivity of the doped ceria samples was measured from 250 °C to 600 °C. A representative impedance spectrum at 250 °C is shown in Figure 7. The various samples were fit using an equivalent circuit with an inductor and two pairs of resistors and constant phase elements (CPEs) in parallel. The conductivity resulting from the electrode was removed during the fitting, and only the conductivity based on the grain and grain boundary were fit with the respective resistor and CPE pairs.

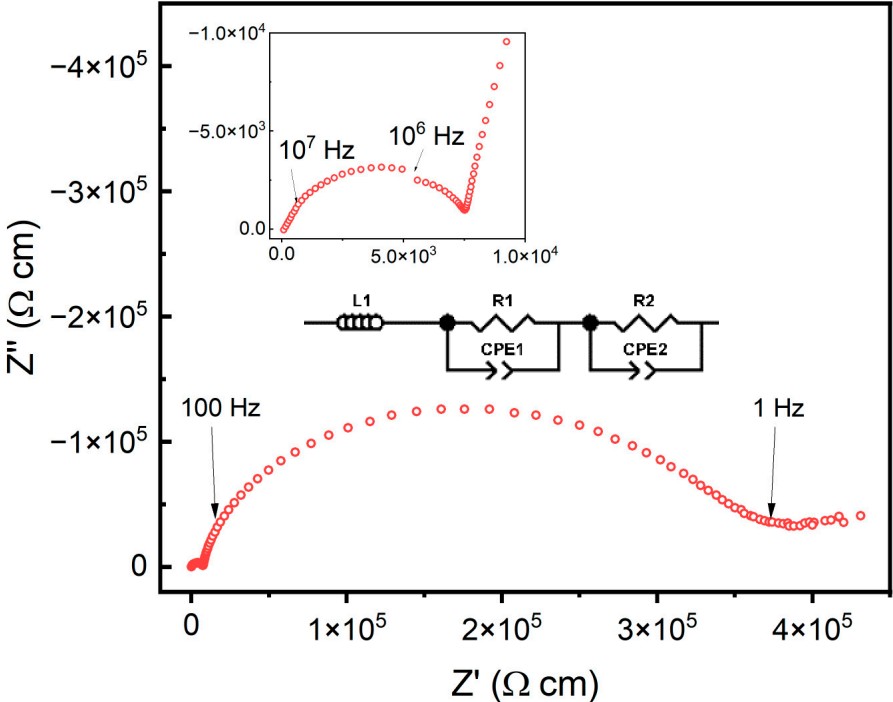

**Figure 7.** Impedance spectra for the 5% doped sample at 250 °C as a representative spectrum for all obtained data. The inset shows the impedance spectra corresponding to the grain. All samples were fit using an equivalent circuit made up of an inductor and two pairs of a resistor and constant phase element in parallel.

The Arrhenius plots for grain and total ionic conductivity for the $Ce_{1-x}(Nd_{x/4}Pr_{x/4}Sm_{x/4}Gd_{x/4})O_{2-\delta}$ samples as a function of temperature are displayed in Figure 8. A 10% gadolinium-doped ceria (GDC10) sample from Omar et al. is used as a reference sample to compare to the results of the multication ceria samples [37]. Values for the activation energy and conductivity are recorded in Table 5. Similar grain conductivity values are observed for all samples at high temperature, but significant differences are observed at low temperature between the different samples. At low temperature, the multication-doped ceria samples perform worse than the singly doped GDC10 sample. The 20% doped sample only displays a conductivity of $1.38 \times 10^{-5}$ S/cm compared to $6.54 \times 10^{-5}$ S/cm for the GDC10 sample. A transition point for the activation energy is observed around 450 °C for all the multication-doped ceria samples. The temperature corresponds to a change in the oxygen ion mobility resulting from an order–disorder or associated–dissociated transition [36,37]. Below the transition temperature, the activation energy is equal to the sum of the migration enthalpy ($\Delta H_m$) and defect association enthalpy ($\Delta H_a$), while above the transition temperature, the activation energy is equal to $\Delta H_m$ [38]. Above the transition temperature, a decrease in the activation energy for the 10%, 15%, and 20% samples is observed due to the dissociation of defect clusters. Above the transition temperature the activation energy is calculated to decrease with increasing dopant concentration, whereas below 450 °C, the activation energy increases with increasing dopant concentration. A similar result was observed by Omar et al. for the co-doped Nd/Sm system as a result of defect cluster formation in the

sample with increasing trivalent cations in the system [37]. Above 450 °C, the activation energy decreases with increasing dopant concentration. The transition from increasing to decreasing may be a result of the difference in entropy within the systems. Above the order–disorder transition temperature, the randomness introduced using doping with four different cations may lead to easier migration of the oxygen ions [39,40]. Above 450 °C, the activation energy for the grain conductivity is on par with singly doped GDC10.

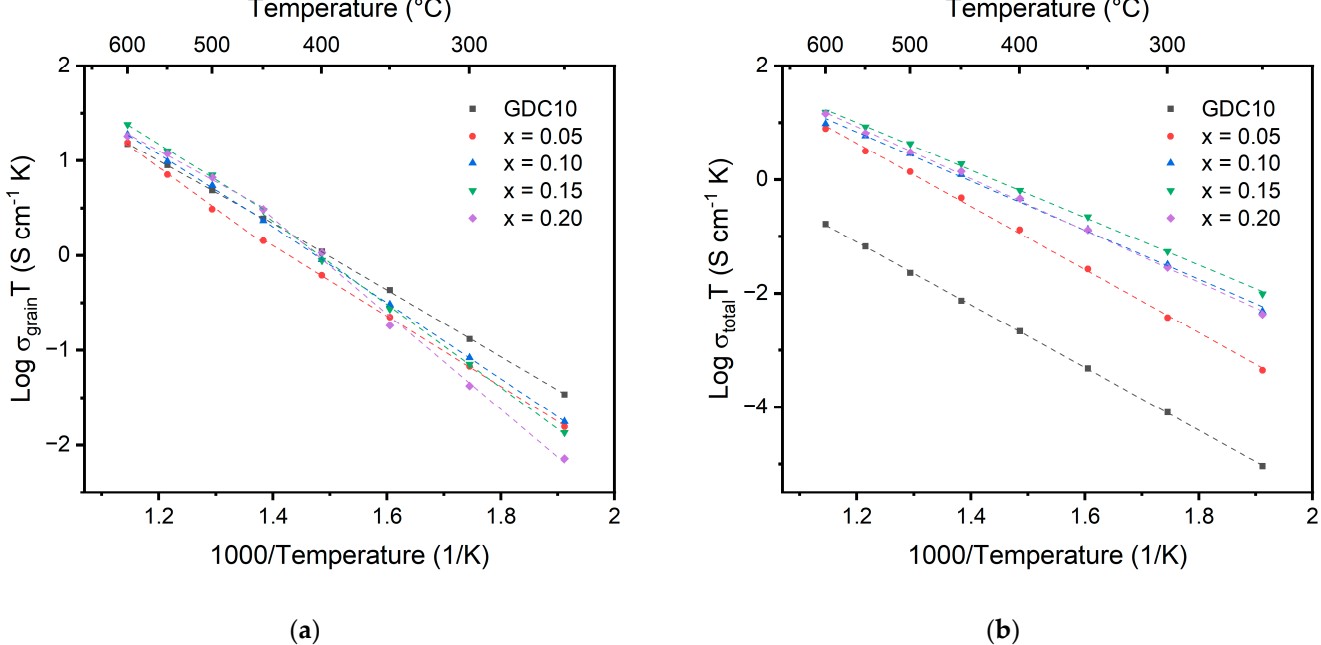

<div align="center">(<b>a</b>)                            (<b>b</b>)</div>

**Figure 8.** Arrhenius plots showing the (**a**) grain and (**b**) total conductivity for doped cerium oxide measured from 250 °C to 600 °C. Data from Omar et al. are included in the figure for the comparison between multication-doped ceria and singly doped ceria (GDC10) [37].

**Table 5.** Activation energy ($E_a$) and conductivity ($\sigma$) values for the doped ceria compounds. Data from Omar et al. are included in the table for the comparison between multication-doped ceria and singly doped ceria (GDC10) [37]. Activation energies below and above the 450 °C transition point are given for the grain conductivity.

| Sample | $E_a$ < 450 °C Grain (eV) | $E_a$ > 450 °C Grain (eV) | $E_a$ Total (eV) | $\sigma_{grain}$ 250 °C (S cm$^{-1}$) ($10^{-5}$) | $\sigma_{total}$ 250 °C (S cm$^{-1}$) ($10^{-6}$) | $\sigma_{grain}$ 600 °C (S cm$^{-1}$) ($10^{-2}$) | $\sigma_{total}$ 600 °C (S cm$^{-1}$) ($10^{-2}$) |
|---|---|---|---|---|---|---|---|
| x = 0.05 | 0.74 | 0.87 | 1.10 | 2.98 | 0.87 | 1.76 | 0.89 |
| x = 0.10 | 0.80 | 0.75 | 0.86 | 3.36 | 9.30 | 2.16 | 1.09 |
| x = 0.15 | 0.87 | 0.74 | 0.83 | 2.58 | 18.7 | 2.76 | 1.75 |
| x = 0.20 | 1.00 | 0.65 | 0.90 | 1.38 | 8.17 | 2.08 | 1.66 |
| GDC10 | 0.65 | 0.60 | 1.03 | 6.54 | 0.20 | 1.72 | 0.02 |

The activation energy values for the total conductivity, as recorded in Table 5, show that the 5% doped sample has the largest total activation energy at 1.01 eV and is on par with previous GDC10 samples. The total activation energy decreases to a minimum of 0.83 eV for the 15% sample. Significant lowering of the activation energy for the total conductivity is likely due to differences in the grain boundary between the samples. Prior studies investigating the conductivity of doped ceria systems have observed an increase in conductivity with an increase in doping concentration up to around 10% to 20% [41]. An increase in conductivity with increasing doping concentration is expected due to the increase in oxygen vacancies, which increase the number of charge carriers within the material. This can be described using Equations (2) and (3), where $[V_o^{\cdot\cdot}]$ is the concentration

of mobile oxygen vacancies, $N_0$ is the number of oxygen sites per volume, $q_i$ is the charge, and $\mu_i$ is the mobility of oxygen ions.

$$RE_2O_3 \xrightarrow[CeO_2]{} 2RE'_{Ce} + 3O_O^X + V_O^{\cdot\cdot} \tag{2}$$

$$\sigma_i = [V_O^{\cdot\cdot}]N_0q_iu_i \tag{3}$$

The ionic conductivity is the product of the concentration, mobility, and charge carriers. In this case, the ionic conductivity occurs via the oxygen vacancies, and for every two trivalent substitutions of $Ce^{4+}$, one oxygen vacancy is formed [5]. Therefore, as the trivalent dopant concentration increases, the ionic conductivity is expected to increase. However, upon increasing the doping level beyond 10% to 20%, the conductivity is observed to decrease even though the number of oxygen vacancies increased. The resulting stable local defect structures of complex defect associates, which form with increasing doping concentration, restrict the motion of oxygen through the lattice and thus decrease the conductivity [9]. A similar trend is observed for the total conductivity of the multication-doped ceria samples. The total conductivity is observed to increase from 5% to 15% dopant concentration but decreases upon further doping to 20%. At 250 °C, a significant benefit is observed for increasing the doping concentration. The 5% doped sample shows a conductivity of only $8.7 \times 10^{-7}$ S/cm compared to the 15% sample at $1.87 \times 10^{-5}$ S/cm, greater than an order of magnitude increase for the 15% doped sample. However, at higher temperatures, the conductivity values for the samples begin to converge due to differences in activation energy. The 15% doped sample remains the highest of the multication-doped samples. The multication-doped samples also maintain at least an order of magnitude greater total conductivity compared to the GDC10 sample, even though the grain conductivities are very similar.

Future work will aim to elucidate the difference in the grain boundary conductivity of the compositions. Due to the large difference observed in the total conductivity compared to singly doped GDC, it is important to further investigate the grain boundaries in the materials, including the presence of nanodomains and local coordination. Nanodomains were previously identified for doped ceria systems even at doping levels less than 20%. Yan et al. specifically used transmission electron microscopy (TEM) with selective area electron diffraction (SAED) to identify nanodomains of the C-type structure [42]. Probing the local configuration of the metal cations and oxygen anions is important to understand how the local order and coordination change with the introduction of more dopants both above and below the transition temperature observed for the activation energy. Scavini et al. and Coduri et al. both used synchrotron XRD with pair distribution function (PDF) to show that while ceria doped with less than 20% of other rare earth elements does maintain a long-range fluorite structure, at length scales < 10 Å, the trivalent dopants preserve their C-type structure [43,44]. Additionally, understanding the composition and preferential segregation of the different cations will be important to understand for the future design of multication-doped ceria compositions. Electron energy-loss spectroscopy (EELS) was used by Bowman et al. to differentiate between Gd and Pr grain boundary segregation in a co-doped ceria material to show that preferential segregation of Pr occurred in the material [45].

### 3. Materials and Methods

*3.1. Synthesis*

Doped cerium oxide, $Ce_{1-x}(Pr_{x/4}Nd_{x/4}Sm_{x/4}Gd_{x/4})O_{2-\delta}$ with $0.05 \leq x \leq 0.20$, was synthesized using the oxalate co-precipitation method. Cerium (III) nitrate hexahydrate (99.9%, Strem Chemicals, Newburyport, MA, USA), praseodymium (III) nitrate hexahydrate (99.9%, Sigma-Aldrich, St. Louis, MO USA), neodymium (III) nitrate hexahydrate (99.9%, Strem Chemicals, Newburyport, MA, USA), samarium (III) nitrate hexahydrate (99.999%, Sigma-Aldrich, St. Louis, MO, USA), gadolinium (III) nitrate hexahydrate (99.99%, Thermo Scientific Chemicals, Waltham, MA, USA), and anhydrous oxalic acid (98%, Thermo Scien-

tific Chemicals, Waltham, MA, USA) were used as precursors. Aqueous solutions of the nitrates and oxalic acid were made in 0.2 M concentrations. A 5% excess concentration of oxalic acid was used to ensure a complete reaction with the metal cations. The two solutions were individually stirred for 15 min to ensure the complete dispersion of the precursors. Then, the nitrate solution was added to the oxalic acid solution, dropwise under vigorous stirring. The solution was aged for 15 min and subsequently washed with deionized water and centrifuged three times. The oxalate powders were then dried in an oven at 50 °C for 24 h. The dried oxalate precursor was removed from the oven and ground using a mortar and pestle before calcination.

Calcination was performed at 800 °C for 2 h (Lindberg Blue M, Lindberg/MPH, Riverside, MI, USA). After calcination, the powders were uniaxially pressed into pellets using 1 wt% PVA as binder under a pressure of 150 MPa. The pellets were pressed using an 8 mm die. The green ceramic pellets were then heat treated to a burnout using a 100 °C/h ramp rate and hold temperatures at 250 °C for 2 h, 450 °C for 2 h, and 650 °C for 4 h. Following burnout, the pellets were sintered at 1450 °C for 8 h. All pellets had a measured geometrical density greater than 93%. Pellets for ionic conductivity were polished to a 1 μm finish using an alumina slurry. Platinum electrodes were then painted onto the surface of the pellet using platinum paste (CL11-5349, Heraeus, Hanau, Germany) and heated to 900 °C for 1 h. Platinum wires were attached to the electrode using silver paint (Ted Pella, Redding, MT, USA) and dried at room temperature.

### 3.2. Characterization Techniques

Simultaneous DSC/TGA (SDT), measuring the change in weight (TGA) and heat flow (DSC), was performed on the oxalate precursors using a TA Instruments Q600. The samples were heated from 30 °C to 800 °C at a 20 °C/min ramp rate in alumina crucibles using flowing air at 100 mL/min. Approximately 10 mg was used for each sample. Onset temperatures were determined using ASTM D3418. X-ray diffraction (XRD) was performed on the oxalate precursor, calcined powders, and sintered pellets using a Panalytical XPert Powder diffractometer. The analysis was performed using a Cu X-ray source with a voltage of 45 kV, step size of 0.016°, dwell time of 15 s, current of 40 mA, and 2θ value of 10–90°. The lattice parameters were determined using GSAS-II software with a Rwp < 15%. The average crystallite size of the powders was estimated using Scherrer's equation from peaks fit using OriginPro 2022b software. Raman spectroscopy was performed using a Horiba microRaman with a 532 nm laser. Spectrums were taken from 100 $cm^{-1}$ to 2000 $cm^{-1}$ for oxalate precursors and from 100 $cm^{-1}$ to 800 $cm^{-1}$ for calcined powders. A D1 filter (10% transparency), 1800 g/mm grating, 100 μm confocal hole, and 10× objective lens were used. Fourier-transform infrared spectroscopy (FTIR) was performed using a Thermofisher Nicolet iS50. The samples were prepared using the KBr pellet method with 250 mg of KBr and 1 mg of oxalate powder. All measurements were made using a DTGS detector in the mid-IR range of 400 $cm^{-1}$ to 4000 $cm^{-1}$ with 64 scans per sample using a 4 $cm^{-1}$ resolution. The peaks were fit using OriginLab with a Gaussian function. Scanning electron microscopy (SEM) and energy dispersive spectroscopy (EDS) were performed using a Tescan Mira 3 with EDAX Octane Pro detector. An accelerating voltage of 10 kV was used for SEM imaging and 15 kV for EDS compositional analysis. The grain size of the sintered pellets was calculated using the Heyn method following ASTM E112-13. The ionic conductivity of the samples was measured using impedance spectroscopy. Each sample was heated in a quartz reactor kept in a small furnace. A thermocouple was placed <5 mm from the sample to maintain accurate sample temperature measurements. The complex impedance of the samples was measured using electrochemical impedance spectroscopy (Solartron 1260, Solartron Analytical, Farnborough, UK) from 250 °C to 600 °C at 50 °C increments. The data were fit using ZView software with an equivalent circuit.

## 4. Conclusions

The multicomponent doping of ceria was investigated using four cations simultaneously, Nd, Pr, Sm, and Gd, as a test case for high-entropy ceramics as ion conductors. The samples were synthesized using the oxalate method. Homogeneous co-precipitation was achieved to produce a single-phase oxalate upon precipitation. The oxalate was observed to decompose as a single species into the oxide following behavior similar to that of cerium oxalate. Raman, XRD, and EDS showed that the homogeneity of the sample was preserved through calcination and sintering. The ionic conductivity measurements reveal no significant improvement in grain conductivity compared to singly doped ceria. A significant decrease in the activation energy for grain conductivity was observed above 450 °C compared to below. The activation energy for the total conductivity was observed to decrease with an increase in doping concentration but remained greater than singly doped GDC10. The total conductivity using multication doping compared to GDC10 was observed to be approximately two orders of magnitude greater at 600 °C.

**Author Contributions:** Conceptualization, J.C.N.; methodology, J.C.N. and E.G.; formal analysis, E.G.; resources, J.C.N.; data curation, E.G.; writing—original draft preparation, E.G.; writing—review and editing, E.G. and J.C.N.; project administration, J.C.N. All authors have read and agreed to the published version of the manuscript.

**Funding:** This research received no external funding.

**Data Availability Statement:** Not applicable.

**Acknowledgments:** The authors thank Youli Wang for his assistance with the EIS setup. We also gratefully acknowledge the Nanoscale Research Facility at the University of Florida for characterization equipment and expertise.

**Conflicts of Interest:** The authors declare no conflict of interest.

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
