# Peer review of "Processing, Phase Stability, and Conductivity of Multication-Doped Ceria"

_inorganics, doi:10.3390/inorganics11070299_

Round 1

Reviewer 1 Report

     In the Introduction only benefits of ceria compounds for IT-SOFCs are shown. No concerns on the electronic conduction of ceria compounds at high temperatures, impeding their use as solid electrolytes in SOFCs.

Suggestions/Corrections

- in the abscissa of Figure 1 (degree) instead of (degrees)

- line 211 (1) and not (1.

- line 310 ...results of the multication ceria samples to [35]?

- line 357 presence, not precense

- line 421 Gaussian, not Guassian

- line 423 was used, not was using

- some references quote one author and et al.; others have the name of all authors.

- in the references care was not taken for super-index and sub-index of the compounds.

Usually High Entropy Ceramics are produced by quenching from high temperature to keep the thermodynamic status of all cations. Here a sintering procedure of 1450 C/ 8 h is followed. How to assure the sintered pellet is a HEC?

Reviewer 2 Report

The article (by Elizabeth Gager  and Juan C. Nino) presents the information about the electrical properties of multicomponent doping of ceria. The crystal and local structure are also discussed. The data are presented in tables and figures; in general, the article is well illustrated.

The presentation of the material in the introduction is well structured. The authors quite fully, but at the same time briefly outlined the main material on doped cerium oxide. The authors define the research problem, they show the importance of this study.

Abstract and conclusions quite fully generalize the presented material.

The article can be accepted for publication after a minor revision.

            The other remarks:

1.      The authors write (line 247): “Additional peaks corresponding to oxygen vacancies can be seen in  doped ceria systems at 250 cm-1, 540 cm-1, and 600 cm-1.” And then “a peak observed around 555 cm-1 is observed for all the doped samples corresponding  to oxygen vacancies” These statements need to be clarified.

The appearance of an oxygen vacancy leads to a decrease in the coordination number of cerium - yes.

Does this result in a shift in F2g  or an appearance of new peak at higher frequencies?

In other words, does lattice compression upon doping lead to a mode F2g shift ? Why the authors believe that this is the result of the formation of an oxygen vacancy.

At the same time, in the caption to Fig.5 the authors write: “A red shift in the F2g peak is observed with increasing dopant concentration as a result of the increase in lattice strain.”

This part of the discussion needs to be clearly structured.

 2.      The authors write (line 321): “above the transition temperature, the activation energy is equal to ΔHa”.

Should be ΔHm.

3.      Since the influence of concentration on conductivity is discussed, it is also expedient to present the concentration dependences of conductivity for clarity.

4.      Information on the relative density of the ceramic samples should be added.
